# Expression and In Vitro Function of Anti-Breast Cancer Llama-Based Single Domain Antibody VHH Expressed in Tobacco Plants

**DOI:** 10.3390/ijms21041354

**Published:** 2020-02-17

**Authors:** Se Ra Park, Jeong-Hwan Lee, Kibum Kim, Taek Min Kim, Seung Ho Lee, Young-Kug Choo, Kyung Soo Kim, Kisung Ko

**Affiliations:** 1Department of Medicine, Medical Research Institute, College of Medicine, Chung-Ang University, Seoul 156-756, Korea; stojr1988@naver.com (S.R.P.); jhlcau@gmail.com (J.-H.L.); kibumkim92@naver.com (K.K.); 2Department of Otorhinolaryngology-Head and Neck Surgery, Chung-Ang University College of Medicine, Seoul 156-756, Korea; 3Major of Nano-Bioengineering, College of Life Sciences and Bioengineering, Incheon-National University, Incheon 22012, Korea; wtrm3977@naver.com (T.M.K.); seungho@inu.ac.kr (S.H.L.); 4Department of Biological science, College of Natural Sciences, Wonkwang University, 460, Iksan-daero, Iksan-si, Jeollabuk-do 54538, Korea; ykchoo@wku.ac.kr

**Keywords:** breast cancer, HER2, Herceptin, VHH antibody, plant antibody

## Abstract

Overexpression of human epidermal growth factor receptor type 2 (HER2) is considered as a prognostic factor of breast cancer, which is positively associated with recurrence when cancer metastasizes to the lymph nodes. Here, we expressed the single variable domain on a heavy chain (VHH) form of anti-HER2 camelid single domain antibody in tobacco plants and compared its in vitro anticancer activities with the anti-HER2 full size antibody. The gene expression cassette containing anti-HER2 camelid single domain antibody VHH fused to human IgG Fc region with KDEL endoplasmic reticulum (ER) (VHH-FcK) was transferred into the tobacco plant via the *Agrobacterium*-mediated transformation. The transformants were screened with polymerase chain reaction and Western blot analyses. Enzyme-linked immunosorbent assay (ELISA) confirmed the binding of the purified anti-HER2 VHH-FcK to the HER2-positive breast cancer cell line, SK-BR-3. Migration assay results confirmed anticancer activity of the plant-derived anticancer camelid single chain antibody. Taken together, we confirmed the possibility of using anti-HER2 VHH-FcK as a therapeutic anticancer agent, which can be expressed and assembled and purified from a plant expression system as an alternative antibody production system.

## 1. Introduction

Breast cancer is associated with high incidence, mortality, and economic and social burden [1]. It is characterized by uncontrolled proliferation of cells that attain the malignant phenotype and is the most common cancer type among women worldwide [2]. Following the discovery of erythroblastic oncogene B2 (*ERBB2*) gene amplification in breast cancer, the human epidermal growth factor receptor 2 (HER2) has emerged as an attractive target for antibody therapies [3]. The protumorigenic properties of HER2, such as strong catalytic kinase activity, extracellular accessibility, high expression, and association with poor prognosis, suggest that it is a promising target for antibody-based therapy [4,5]. Several therapeutic methods have been developed to block HER2 activity, thereby suppressing tumor growth, including the use of mAbs such as trastuzumab [6]. 

Although applications of therapeutic mAbs have been increasing, the current mammalian-based systems of antibody production produce only limited quantities at high costs [7]. Other available systems such as bacteria and yeasts lack the specific machinery for post-translational modifications of proteins that are essential for the production of functional mAbs [8]. Plants offer several advantages for the mAb expression, such as lack of human pathogenic contaminants, cost-effective cultivation methods, low scale-up costs, and glycosylation efficiency [9]. Transgenic plants allow for stable gene insertion and easy propagation through in vitro tissue culture and seedlings [10], and thus have emerged as efficient systems for the antibody production [11,12]. 

Various forms of antibodies, including full size, ScFv, minibody, and camelid, have been modified and expressed in heterologous biosystems [13]. Camelid antibodies with two identical heavy chains (VHH) are fully capable of binding antigens in the absence of light chains. These VHHs offer several advantages for biotechnological applications [13]. The VHHs have unique characteristics such as size, intrinsic stability, ease of production, and potential for therapy and diagnosis [5,14,15,16]. Thus, the hypothesis of this current study is that the llama-based anti-HER2 VHH-Fc antibody can be expressed in plants with its similar anticancer activities to the commercial full size mAb. However, no previous study has yet been conducted for the expression and commercialization of Rama-based anti-HER2 antibody in plants.

In the present study, to test the hypothesis, an anti-HER2 VHH antibody fused to the Fc fragment was expressed in plants, and its anticancer activity was compared with that of the parental full-sized anti-HER2 mAb.

Antibody-dependent cellular cytotoxicity (ADCC) is mediated by immune cells expressing Fc gamma receptors in response to the binding of the antibody to a tumor or viral antigen. Thus, the Fc domain is essential for ADCC induction.

## 2. Results

### 2.1. Generation of Tobacco Transgenic Plants Expressing Anti-HER2 VHH-FcK

Polymerase chain reaction (PCR) analysis confirmed the expression of the transgene encoding anti-HER2 VHH-FcK in the in vitro regenerated tobacco plants. The anti-HER2 VHH-FcK gene was detected in the regenerants with PCR amplification using specific primers. The expected band size (1176 bp) for anti-HER2 VHH-FcK gene was observed in the transformants (Figure 1B). No DNA product was detected in nontransgenic plants. The anti-HER2 VHH-FcK gene fragment was amplified using the plasmid pBI121 carrying the expression cassette of anti-HER2 VHH-FcK as a positive control [17]. Western blot analysis using horseradish peroxidase (HRP)-conjugated antihuman Fc antibody verified the expression of anti-HER2 VHH-FcK in leaf extracts. A band corresponding to the molecular weight of Herceptin (positive control) was observed at approximately 50 kDa, whereas a band corresponding to that of plant–derived anti-HER2 VHH-FcK (anti-HER2 VHH-FcK^P^) was detected at 44 kDa. The band was not detected in nontransgenic plants (Figure 1C). The in vitro transgenic plants with anti-HER2 VHH-FcK expression were grown in soil pots under greenhouse conditions.

### 2.2. Binding Activity of the Plant-Derived Anti-HER2 VHH-FcK to Human Breast Cancer Cells

To test the binding activity of the plant-derived anti-HER2 VHH-FcK to human breast cancer cells, the recombinant protein was purified from leaf biomass. Sodium dodecyl sulfate polyacrylamide gel electrophoresis (SDS-PAGE) results confirmed the successful purification of anti-HER2 VHH-FcK^P^ using Protein A affinity chromatography (Amicogen, Jinju, Korea) (Figure 2). The results of enzyme-linked immunosorbent assay (ELISA) showed strong anti-HER2 VHH-FcK^P^ signals for SK-BR-3 cells (Figure 2 and Figure 3). The signals obtained for MDA-MB-231, MCF-F, and SW480 cells were positive, but not strong. Anti-HER2 VHH-FcK from tobacco plants bound to the cells of the human breast cancer cell line, SK-BR-3 (Figure 2 and Figure 3).

We also used ELISA to compare the binding activity of Herceptin (positive control) and anti-HER2 VHH-FcK to SK-BR-3 breast cancer cells (HER2-positive) (Figure 3). When the dilution factor was 2 or 1, no significant difference in the binding activity was observed between the two antibodies. However, when the dilution factor was 0.5, 0.25, 0.125, 0.0625, or 0.03125, the absorbance value was slightly lower for anti-HER2 VHH-FcK than for Herceptin (Figure 3). To compare the specific binding activity of anti-HER2 VHH-FcK to various breast cancer cells, ELISA was conducted using SK-BR-3 (HER2-positive), MCF-7 (HER2-negative), and MDA-MB-231 (HER2-negative) cells (Figure 4).

In SK-BR-3 cells, both anti-HER2 VHH-FcK^P^ and Herceptin showed an absorbance value of more than 1.2. However, the value was less than 0.5 for MCF-7 cells. In MDA-MB-231 cells, Herceptin and anti-HER2 VHH-FcK^P^ showed an absorbance of 0.2 and 0.5, respectively. In SW480 (HER2-negative) cells, the absorbance value was ~0.5 and ~0.6 for Herceptin and anti-HER2 VHH-FcK^P^, respectively (Figure 4). We used 1× phosphate-buffered saline (PBS) as a negative control to obtain a basal value of 0.1 for all cell lines (Figure 3 and Figure 4). 

### 2.3. Cancer Cell Migration Assay

Transwell migration assay was performed to determine the inhibitory activities of the plant-derived anti-HER2 VHH and Herceptin (Figure 5A,B). The number of migratory cells was lower in the plant-derived anti-HER2 VHH treatment group (~30 cells/field) than that in the Herceptin treatment group (positive control; ~50 cells/field) (Figure 5B). The control and nonspecific IgG treatment groups (negative controls) showed a significantly higher number of migratory cells (~70 and ~80 cells/field, respectively) (** *p* < 0.01, * *p* < 0.05) (Figure 5B).

### 2.4. N-glycan Structure of Anti-HER2 VHH-FcK

The glycan structures of anti-HER2 VHH-FcK, RNase B (RB), and horseradish peroxidase (HRP) were analyzed with high-performance liquid chromatography (HPLC; Figure 6). The HPLC profiles of plant originated HRP after PNGase A treatment showed a plant complex-type *N*-glycans, whereas the plant-derived HER2VHH-FcK exhibited only high mannose-type glycan structures (Figure 6). RB showed high mannose-type glycan structures similar to those of anti-HER2 VHH-FcK (Figure 6). The RB and anti-HER2 VHH-FcK glycans were similar peaks from 19 to 24 min with a slight difference. However, the peaks for RB were slightly lower than those for anti-HER2 VHH-FcK.

## 3. Discussion

Here, a llama-single-domain anti-HER2 antibody with KDEL, anti-HER2 VHH-FcK, was expressed in tobacco and analyzed for its anticancer functions. We confirmed insertion and expression of anti-HER2 VHH-FcK gene (1176 bp) in tobacco plants via PCR and immunoblot analysis. In addition, SDS-PAGE results showed the successful purification of anti-HER2 VHH-FcK^P^. The antibody binding activity to breast cancer cell expressing HER2 was identified by cell ELISA analysis. The absorbance value of anti-HER2 VHH-FcK^P^ was similar to Herceptin in the SK-BR-3 breast cancer line. Moreover, ELISA analysis in various cancer cell lines showed a slight difference in specific binding activity of llama-based plant-derived anti-HER2 VHH FcK to breast cancer lines compared with Herceptin. From this point of view, the plant-derived anti-HER2 VHH-FcK antibody (anti-HER2 VHH-FcK^P^) showed specific binding to HER2-positive tumor cells, consistent with the observation for Herceptin (positive control). The binding of anti-HER2 VHH-FcK to HER2 almost matched that of Herceptin, and anti-HER2 VHH-FcK showed higher affinity for tumor cell in vitro; this is suggestive of its potential as an alternative therapeutic approach for HER2-positive tumors [18]. 

Post-translational modification of recombinant proteins essentially determines their immunogenicity. In eukaryotic cells, glycoproteins are *N*-glycosylated in two distinct organelles, namely, the endoplasmic reticulum (ER) and Golgi bodies [19,20]. Glycan structures significantly affect the key characteristics of therapeutic proteins, such as folding, in vivo half-life, and immunity [21]. As the *N*-glycan modification in plants differs from that in mammals, the recombinant proteins produced in transgenic plants are not identical to mammalian glycoproteins [22]. Further, the folding and assembly of recombinant proteins, including the transfer of oligosaccharide precursors to the *N*-glycosylation sites, may be precisely accomplished in transgenic plant systems [23]. However, plant-specific glycan residues α(1,3)-xylose and β-fucose may induce allergic reactions upon administration to humans [22]. This limitation may be solved by the using KDEL (ER retention sequence) to retain the glycoproteins in the ER to avoid plant-specific glycan attachment in the plant during the production of therapeutic glycoproteins [22,24,25,26]. In the present study, the *N*-glycan structure of anti-HER2 VHH-FcK^P^ was a high mannose type owing to the KDEL sequence. This result is supported by our previous studies where the KDEL-fused proteins showed mainly high mannose type glycan structure [21,22].

In the migration assay, the number of migratory cells was significantly lower in anti-HER2 VHH-FcK^P^ treatment group than that in the Herceptin treatment group (positive control). This observation suggests that the plant-derived llama-anticancer large single chain antibody could be properly expressed and assembled, and that it exhibited anticancer activity. 

In plants, the expression of a full-sized human antibody with heavy and light chains necessitates the use of two promoters to regulate the expression of each chain [27,28]. In general, it is difficult to obtain similar expression levels for both heavy and light chains. However, the expression of a camelid-based antibody with a single-domain form may be carried out using a single promoter [29]. Camelid-based antibodies offer advantages such as ease of obtaining fully functional multiple antibodies from a single plant cell without worrying about the expression of both heavy and light chains. Our previous study showed that the heavy and light chains from two different full size antibodies expressed in a single plant could not be properly assembled together with their own heavy or light chain, eventually resulting in the formation of chimeric antibodies with reduced activities [30]. In contrast, multiple camelid-based antibodies could be expressed in a single plant cell without such chimerism.

In conclusion, a single llama-based anti-HER2 VHH-FcK was successfully expressed in tobacco, and it had specific binding activity to breast cancer cells expressing HER2 with inhibition activity on breast cancer cell migration. The efficient antitumor effects of the plant-derived anti-HER2 VHH-FcK—similar to those of the positive control Herceptin—may extend its application as an alternative approach to combat HER2-overexpressing tumors [18]. 

## 4. Materials and Methods

### 4.1. Construction of Plant Expression Vector

The cDNA encoding anti-HER2 VHH fragment fused to the Fc carrying KDEL ER retention motif (anti-HER2 VHH-FcK) was cloned into pBHA vector (Bioneer, Daejeon, Korea). The recombinant vector was introduced into DH5α competent cells and isolated using Plus Plasmid Mini Kit (Favorgen, Ping-Tung, Taiwan). The vector was digested by restriction enzymes (*Nco*I and *Bam*HI) to insert anti-HER2 VHH-FcK encoding genes. The anti-HER2 VHH-FcK gene was cloned under the control of the cauliflower mosaic virus 35S promoter with duplicated upstream B domains (Ca2 promoter) and the untranslated leader sequence of alfalfa mosaic virus RNA4 [31] into the plant expression vector pBI121 (Figure 1) [32].

### 4.2. Plant Transformation

The recombinant plant expression vector was transferred into *Agrobacterium tumefaciens* strain LBA4404 by electroporation. Transgenic tobacco (*Nicotiana tabacum* Xanthi) plants were generated by *Agrobacterium*-mediated transformation [33]. The in vitro cultivated tobacco leaves were cut in two or three pieces and incubated with Murashige and Skoog (MS) medium for 1 min with gentle shaking. The leaves obtained from the bacterial suspension culture were used as explants and placed upside down on Petri dishes containing a cocultivation medium supplemented with sucrose and agar without antibiotics. After 2 days in culture, the explants were transferred onto a regeneration medium with cefotaxime (250 mg/L) and kanamycin (100 mg/L). The dishes were incubated in a growth chamber at 25 °C, and the medium was changed every week. Regenerated shoots were separated from the callus on the explant and cultured on MS medium (Dachfu, Haarlem, Netherland) with kanamycin (100 mg/L) [31]. Transgenic tobacco lines were selected on MS medium with kanamycin (100 mg/L). The transgenic plants were transferred to a greenhouse to obtain whole plants. Transgenic and nontransgenic plants were cultivated in greenhouse under controlled conditions [33].

### 4.3. DNA Isolation and PCR

Genomic DNA was isolated from tobacco leaves using HiYield^TM^ genomic DNA mini kit (RBC Bioscience, Seoul, Korea), as per the manufacturer’s recommendations. PCR was performed to confirm the insertion of anti-HER2 VHH-FcK genes for the selection of the transformants. The primers used were 5′-ACTTCCACCATGGCTACTCAACG-3′ (forward) and 5′-GGATCCTCTAGATCAGA GTTCATCTTT-3′ (reverse). PCR was subjected to 30 cycles of 95 °C for 20 s, 60 °C for 20 s, and 72 °C for 60 s. The leaves from the nontransgenic tobacco plant were used as a negative control, while pBI121 carrying the anti-HER2 VHH-FcK expression cassette was used as a positive control. 

### 4.4. Western Blot Analysis

Leaf samples freshly obtained (100 mg) were ground in 300 μL of 1× PBS (10 mM disodium phosphate (Na_2_HPO_4_), 2.7 mM potassium chloride (KCl), 137 mM sodium chloride (NaCl), and 2 mM potassium phosphate (KH_2_PO_4_)). The homogenized sample 20 μL was mixed with 4 μL of a loading buffer (5% 2-mercaptoethanol, 50% glycerol, 100 mM Tris-HCl, 10% SDS, and 0.1% bromophenol blue) and loaded onto a 12% SDS-PAGE gel. The proteins were electrophoresed using 1× SDS running buffer (0.2 M glycine, 25 mM Tris-HCl, 0.1% (*w*/*v*) SDS), and the separated bands were transferred onto a nitrocellulose membrane for 90 min at 4 °C. The membrane was soaked in a blocking buffer (5% skimmed milk in 1× PBS) for 1 h at room temperature and probed with an HRP-conjugated antibody (goat antimurine antibody; Jackson Immuno Research, West Grove, PA, USA) for 90 min at room temperature with gentle shaking. The membrane was washed thrice (10 min per wash) with a washing buffer (1× PBS containing 0.001% Tween-20) and treated with Clarity^TM^ Western ECL substrate (Bio-Rad Labs, Hercules, CA, USA) for detection [29].

### 4.5. Plant Growth and Purification

To purify the plant-derived anti-HER2 VHH-FcK, 200 g of leaves were homogenized with extraction buffer (37.5 mM Tris/HCl pH 7.5, 50 mM NaCl, 15 mM ethylenediaminetetra acetic acid (EDTA) pH 8.0, 75 mM sodium citrate, and 0.2% sodium thiosulfate) and centrifuged at 9000× *g* for 30 min at 4 °C [34]. The supernatant was filtered using Miracloth (Biosciences, La Jolla, CA, USA). The protein solution, adjusted to pH 5.1, was centrifuged for 30 min at 10,000× *g*, and the supernatant was filtered using Miracloth. The pH was adjusted to 7.0 using 3000 mM Tris-HCl, and the supernatant was treated with 8% ammonium sulfate. After 2 h incubation at 4 °C, the solution was centrifuged at 4 °C for 30 min at 9000× *g*, and the precipitate discarded. Ammonium sulfate was added at a final concentration of 22.6%, and the solution was incubated for 16 h at 4 °C. The pellet was resuspended in an extraction buffer (1/10 of the starting extraction buffer volume) and the obtained solution was centrifuged at 4 °C for 30 min at 10,000× *g*. After filtration with Miracloth, Protein A affinity chromatography (Amicogen, Jinju, Korea) was used to purify. Anti-HER2 VHH-FcK^P^ was dialyzed against autoclaved 1× PBS buffer (10 mM Na_2_HPO_4_, 2.7 mM KCl, 137 mM NaCl, and 2 mM KH_2_PO_4_) for further study. The purified protein separated in SDS-PAGE gel as destained above was stained with the anti-HER2 VHH-FcK^P^ protein using Coomassie blue staining solution (30% methanol (*v*/*v*), 10% acetic acid (*v*/*v*), 0.01% Coomassie blue (*w*/*v*)) for 30 min at room temperature with gentle shaking, followed by destaining with 10% acetic acid.

### 4.6. ELISA

The cancer cell binding activity of anti-HER2 VHH-FcK^P^ was confirmed with ELISA. SK-BR-3, MCF-7, and MDA-MB-231 cells (breast cancer cells) were cultured in Roswell Park Memorial Institute (RPMI)-1640 medium (HyClone, South Logan, UT, USA) supplemented with 10% heat-inactivated fetal bovine serum (FBS), penicillin, and streptomycin, and were grown at 37 °C in 5% CO_2_ overnight. Cells (4 × 10^5^ cells/100 μL per well) were seeded into 96-well plates and were fixed with 150 μL/well of 4% paraformaldehyde (PFA) in 1× PBS for 20 min at room temperature. The cells were washed thrice with 1× PBS and blocked with 200 μL/well of 1% bovine serum albumin (BSA) in 1× PBS at 37 °C for 1 h and incubated overnight at 4 °C with 100 μL/well of purified protein. The cells were then probed with HRP-conjugated goat antihuman IgG (1:8000 diluted; Jackson Immuno Research, West Grove, PA, USA) at 37 °C for 2 h and treated with 3,3′,5,5′-tetramethylbenzidine (TMB) substrate (KPL, Gaithersburg, MA, USA) for 10 min for signal detection. The reaction activity was terminated using TMB stop solution (KPL), and measured at 450 nm using an ELISA reader (BioTek, Highland, VT, USA).

### 4.7. Cancer Cell Migration Assay with Anti-HER2 VHH FcK^P^

Cancer cell migration analysis was conducted using Transwell Permeable Supports with 3 μm-pore size (Corning Inc, Corning, NY, USA). The membrane was coated with human laminin mixture (Chemicon, Temecula, CA, USA), rat laminin-5 (Chemicon, Temecula, CA, USA), collagen I, or fibronectin (Sigma, St. Louis, MO, USA) at a concentration of 0.5 μg/mL in 1× PBS, overnight at 4 °C. Fifty thousand cells were seeded in 300 μL medium in the upper chamber and incubated at 37 °C for 6 h in a CO_2_ incubator. The cells migrating to the bottom layer were stained with 5% crystal violet and counted under a microscope. Cell invasion assay was conducted in an extracellular matrix (ECM) invasion chamber (Chemicon, Temecula, CA, USA), where the upper layer of the transmembrane was coated with Matrigel (Corning, Corning, NY, USA). SK-BR-3 human breast cancer cells (3 × 10^5^ cells) were loaded in the upper chamber in serum-free RPMI and preincubated for 10 min at 37 °C in a CO_2_ incubator, followed by additional incubation for 36 h at 37 °C. The cells migrating to the bottom layer were stained for 1 h using 5% crystal violet and counted under a microscope. To determine the effect of anti-HER2 VHH FcK^P^ and Herceptin on cell invasion, the cells were incubated with 20 μg/mL antigoat mAb (Jackson Immuno Research, West Grove, PA, USA) [9].

### 4.8. HPLC Analysis for N-glycan

The purified anti-HER2 VHH-FcK was incubated along with 2 μL of pepsin in an incubator for 16 h at 37 °C for digesting it into glycopeptides. The digested glycopeptides were obtained through a C18 sep-pak cartridge (Waters, Lexington, MA, USA) and washed twice with 1 mL of 5% acetic acid for the removal of contaminants such as salts and free sugar. The fraction containing peptides and glycopeptides was eluted with a series of solutions containing 20% 2-propanol in 5% acetic acid, 4% 2-propanol in 5% acetic acid, and 100% 2-propanol. A speed vacuum was used to dry the eluted fractions and PNGase A (Roche, Basel, Switzerland) was applied to incubate the dried samples for 16 h at 37 °C to release *N*-glycans. The released *N*-glycans were purified using a graphitized carbon resin procured from Carbograph (Alltech, Lexington, MA, USA) [31]. For *N*-glycan detection with HPLC, the purified glycans were aminobenzamide (AB)-labeled by making a slight modification to a previously described method [35]. The labeling reagent was obtained by dissolving 6 mg of AB in 100 μL of 30% (*v*/*v*) acetic acid in a dimethyl sulfoxide solution containing 1000 mM of sodium cyanoborohydride [22]. Each dried glycan sample was dissolved in 5 μL of labeling reagent and incubated for 16 h in a tightly capped tube at 37 °C. The mixture was diluted with 1 mL of acetonitrile (96% (*v*/*v*) in water), and the excess of labeling reagent was removed using a cyano cartridge (Agilent Technologies, Santa Clara, CA, USA). The purified AB-labeled glycans were separated on a TSK amide-80 (5 μm, 4.6 × 250 mm) column (Tosoh Bioscience, Prussia, PA, USA) using an HPLC system with a fluorescence detector (330 nm excitation and 425 nm emission) [21]. Separations were achieved at a flow rate of 1.0 mL/min using a mixture of solution A (100% acetonitrile) and solution B (50 mM ammonium formate). After the column was equilibrated with 30% solution B, samples were injected and eluted using a linear gradient of 45% solution B for 60 min. 

### 4.9. Statistical Analysis

All values are shown as the mean ± SD. Anti-HER2 VHH-FcK^P^ and Herceptin were compared by using the unpaired *t*-test, and *p*-values less than 0.05 (*) and 0.01(**) were considered statistically significant. Statistical significance was assessed using Excel (Microsoft Office Excel; Microsoft Corporation, Redmond, WA, USA). The difference between anti-HER2 VHH-FcK^P^ and Herceptin were compared statistically (* *p* < 0.05, ** *p* < 0.01).

## Figures and Tables

**Figure 1 ijms-21-01354-f001:**
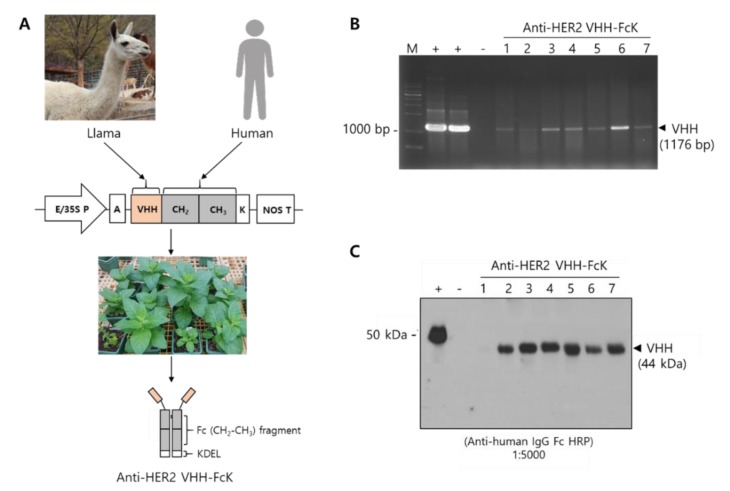
Schematic illustration of production of the plant-derived anti-HER2 VHH-FcK antibody. (**A**) Construction of plant expression for vector VHH camelid human Fc fusion antibody. In vitro transgenic tobacco plants were generated to express anti-HER2 VHH-FcK using *Agrobacterium*-mediated transformation. E/35S P and NOS T represent promoter and terminator, respectively. A: Alfalfa mosaic virus (AMV) leader sequence; K: KDEL (ER retention signal) sequence. (**B**) PCR analysis to confirm the insertion of the transgene encoding VHH-CH_2_-CH_3_ antibody. (**C**) Western blot analysis to confirm anti-HER2 VHH-FcK protein expression. +: positive control, -: negative control.

**Figure 2 ijms-21-01354-f002:**
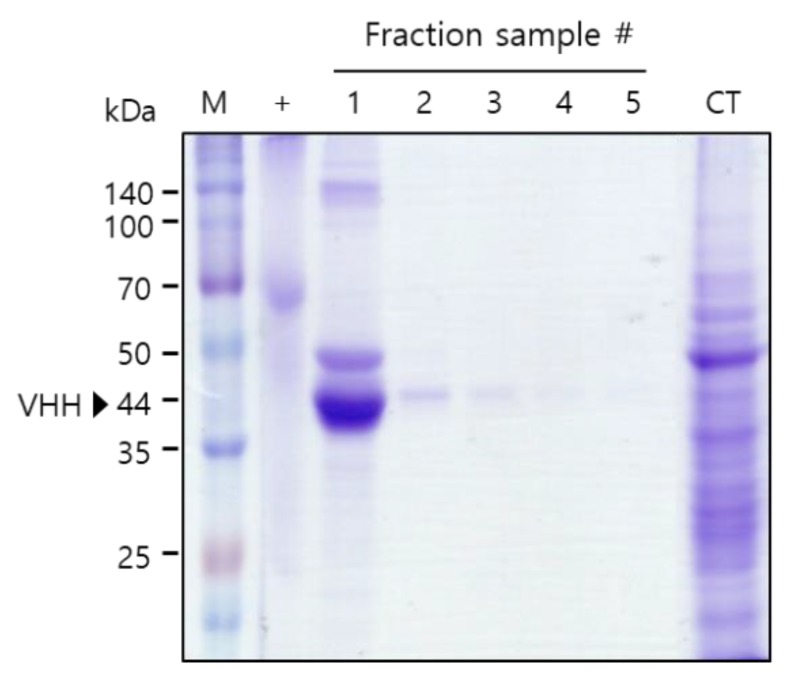
Sodium dodecyl sulfate polyacrylamide gel electrophoresis (SDS-PAGE) to confirm the purification of anti-HER2 VHH-FcK from tobacco leaf. Confirmation of the purified anti-HER2 VHH-FcK^P^ using SDS-PAGE. M: protein marker; +: positive control, BSA (2 μg); 1-5: anti-HER2 VHH-FcK fraction samples; CT: column flowthrough.

**Figure 3 ijms-21-01354-f003:**
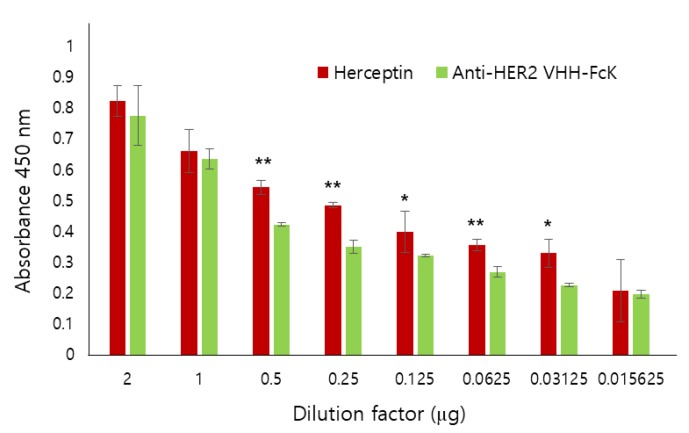
Comparison of the binding activity of Herceptin and the anti-HER2 VHH-FcK^P^ to SK-BR-3 breast cancer cell. Anti-HER2 VHH-FcK and Herceptin, serially diluted from 2 to 0.015625 μg/μL, were applied to ELISA plates seeded with SK-BR-3 cells. We used 1× PBS as a negative control. The absorbance was measured at 450 nm wavelength. Error bars indicate standard deviation (* *p* < 0.05, ** *p* < 0.01).

**Figure 4 ijms-21-01354-f004:**
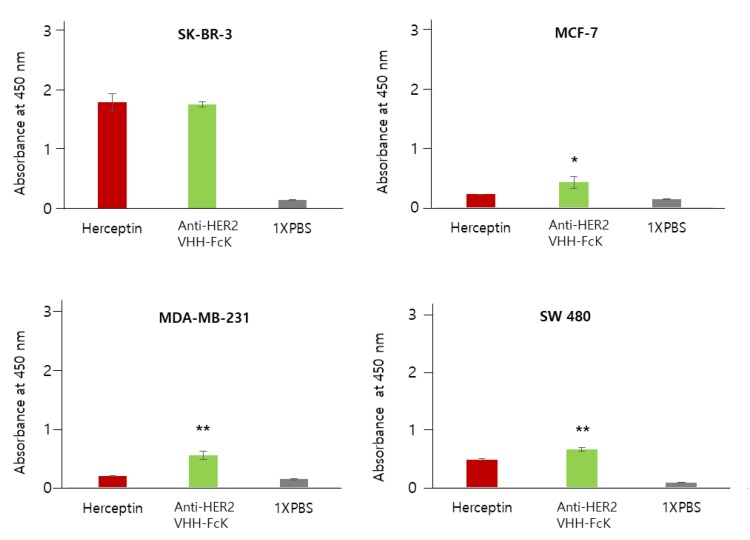
Comparison of the specific binding activity of Herceptin and the anti-HER2 VHH-FcK^P^ to breast cancer cell lines SK-BR-3, MCF-7, and MDA-MB-231, and the colorectal cancer cell line, SW480. Herceptin and plant-derived anti-HER2 VHH-FcK were applied to ELISA 96-well plates seeded with breast cancer cell lines SK-BR-3, MCF-7, and MDA-MB-231 and the colorectal cancer cell line, SW480 (4 × 10^5^ cells/100 μL). The difference between anti-HER2 VHH-FcK^P^ and Herceptin were compared statistically (** *p* < 0.01, * *p* < 0.05).

**Figure 5 ijms-21-01354-f005:**
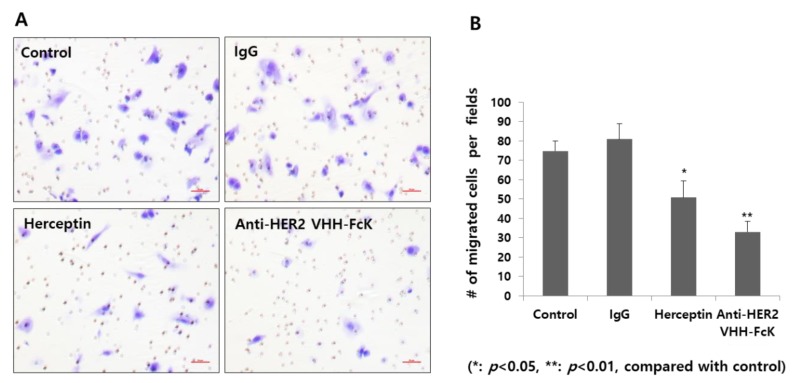
Migration assay for cells treated with the plant-derived anti-HER2 VHH FcK and Herceptin. (**A**) Microscopic observation of the migratory cells treated with 1× PBS (negative control), nonspecific IgG (negative control), Herceptin (positive control), and plant-derived anti-HER2 VHH FcK. (**B**) Quantified cell number per field from microscopic observation (A). Scale bar (red) represents 50 μm (A).

**Figure 6 ijms-21-01354-f006:**
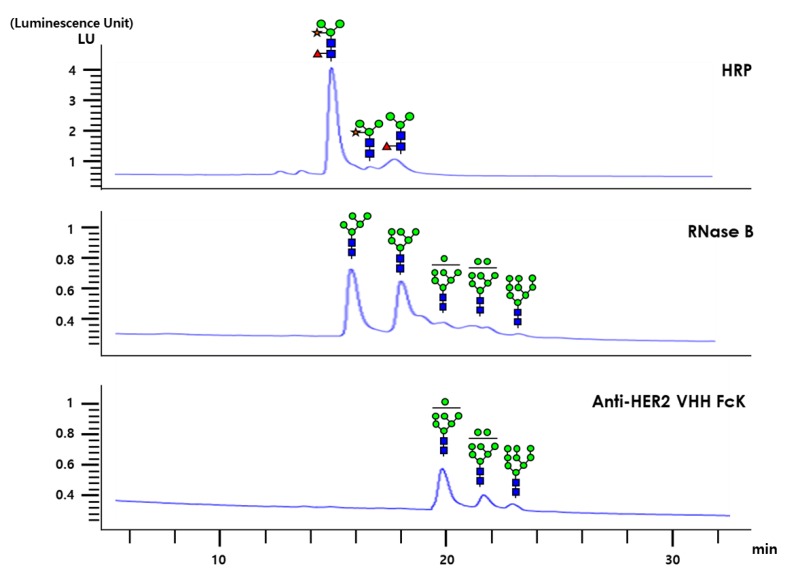
HPLC analysis for glycan structures of bovine pancreas-derived RNase B (RB), plant-derived HRP, and plant-derived anti-HER2 VHH-FcK. The specific glycan structures of different peaks are presented. GlcNAc, fucose, mannose, and xylose are depicted as blue squares, red triangles, green circles, and yellow stars, respectively.

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
