# Peer review of "Expression and In Vitro Function of Anti-Breast Cancer Llama-Based Single Domain Antibody VHH Expressed in Tobacco Plants"

_ijms, 2020, doi:10.3390/ijms21041354_

Round 1
Reviewer 1 Report
Overview: In this manuscript, the authors evaluated an interesting topic of using plant models for mAb expression with anticancer activity. Despite the importance of the topic, but the manuscript contain several concerns need to be addressed to be fit for publication as follows:
Major concerns:
1. The statistical analysis is a must to validate the findings of the study but it is missed here. Thus, a separate section clarifying the model used for statistical analysis, the types of software used should be added. Additionally, statistical analysis should be clearly described in the results section.
2. The discussion needs to be rewritten with more depth and enriched with the comparison to the earlier depth without repetition of the information already present in the introduction.
Specific comments:
1. The title needs to be represented by specified the camelid with "llama" and plant "tobacco".
2. Abstract; in the conclusion add "as a therapeutic anti-cancer agent".
3. Introduction: the hypothesis of the study should be clearly described before the aim.
4. Results: remove all repetition of the methods in the results section. E.g. line 132-134.
5. Abbreviations: the manuscript is full of abbreviations but the authors have chosen only 5 abbreviations.
Author Response
Thank you for considering this manuscript entitled "Expression and in vitro function of anti-breast cancer llama-based single domain antibody VHH expressed in tobacco plants" for publication in IJMS. We are grateful to reviewers for the valuable suggestions. Here are the responses to the comments of reviewers.
Reviewer : 1
In this manuscript, the authors evaluated an interesting topic of using plant models for mAb expression with anticancer activity. Despite the importance of the topic, but the manuscript contain several concerns need to be addressed to be fit for publication as follows:
Major concerns:
Q1. The statistical analysis is a must to validate the findings of the study but it is missed here. Thus, a separate section clarifying the model used for statistical analysis, the types of software used should be added. Additionally, statistical analysis should be clearly described in the results section.
→ A separate section clarifying the model used for statistical analysis is added as follows.
(Page 10 line 350~355) Statistical Analysis
All values are shown as the mean ± SD. Anti-HER2 VHH-FcKP and Herceptin were compared by using the unpaired t-test, and P-values less than 0.05 (∗) and 0.01 (**) were considered statistically significant. Statistical significance was assessed using Excel (Microsoft Office Excel; Microsoft Corporation, Redmond, WA).
“The difference between anti-HER2 VHH-FcKP and Herceptin were compared statistically (*P<0.05, **P <0.01).” is added in Fig.3 and 4.
Q2. The discussion needs to be rewritten with more depth and enriched with the comparison to the earlier depth without repetition of the information already present in the introduction.
Specific comments:
→ The sentence “Plant as an expression system for the production of recombinant therapeutic proteins offers several advantages over other systems—such as animals and microorganisms [18]—including low production cost and easy scalability.” is removed from ‘Discussion’ part since it is redundant according to the reviewer’s suggestion.
In addition, according to the reviewer’s suggestion, several sentences with more depth discussion are included as follows.
The sentences “We confirmed insertion and expression of anti-HER2 VHH-FcK gene (1176 bp) in tobacco plants via PCR and immunoblot analyses. In addition, SDS-PAGE results showed the successful purification of anti-HER2 VHH-FcKP. The antibody binding activity to breast cancer cell expressing HER2 was identified by cell ELISA analysis. The absorbance value of anti-HER2 VHH-FcKP was similar to Herceptin in SK-BR-3 breast cancer cell line. Moreover, ELISA analysis in various cancer cell lines showed slightly difference in specific binding activity of llama-based plant-derived anti-HER2 VHH-FcK to breast cancer lines compared with Herceptin.” are included (Page 7 lines 177~184).
The sentence “This result is supported by our previous studies where the KDEL-fused proteins showed mainly high mannose type glycan structure [21, 22].” is included (Page 7 line 202~204).
The sentence “In conclusion, a single llama-based anti-HER2 VHH-FcK was successfully expressed in tobacco plant, and it had specific binding activity to breast cancer cells expressing HER2 with inhibition activity on breast cancer cell migration.” is newly written (Page 8 line 220~222).
The title needs to be represented by specified the camelid with "llama" and plant "tobacco".
→ The title is changed to “Expression and in vitro function of anti-breast cancer llama-based single domain antibody VHH expressed in tobacco plants” according to the reviewer’s suggestion.
→ In conclusion sentence, “as a therapeutic anti-cancer agent” is added (Page 1 line 28).
Introduction: the hypothesis of the study should be clearly described before the aim.
→ The hypothesis “Thus, the hypothesis of this current study is that the llama-based anti-HER2 VHH-Fc antibody can be expressed in plant with its similar anti-cancer activities to the commercial full size mAb.” is described before the aims (Page 2 line 56~58).
Results: remove all repetition of the methods in the results section. E.g. line 132-134.→ The repeated method description sentences are removed as follows. The sentence “The lower part of the transwell membrane was coated with extracellular matrix components, while the upper chamber was seeded with cells. Cancer cell migration to the bottom membrane was observed under a microscope” (Page 5 line 141-143) is removed.
Abbreviations: the manuscript is full of abbreviations but the authors have chosen only 5 abbreviations.
→ All abbreviations are included as follows.
|
AB |
Aminobenzamide |
|
ADCC |
Antibody-dependent cellular cytotoxicity |
|
AMV |
Alfalfa mosaic virus |
|
BSA |
Bovine serum albumin |
|
ECM |
Extracellular matrix |
|
EDTA |
Ethylenediaminetetra acetic acid |
|
ELISA |
Enzyme-linked immunosorbent assay |
|
ER |
Endoplasmic reticulum |
|
ERBB2 |
Erythroblastic oncogene B2 |
|
FBS |
Fetal bovine serum |
|
HER2 |
Human epidermal growth factor receptor type 2 |
|
ERBB2 |
Erythroblastic oncogene B2 |
|
HPLC |
High-performance liquid chromatography |
|
HRP MS |
Horseradish peroxidase Murashige and skoog |
|
PBS PCR |
Phosphate-buffered saline Polymerase chain reaction |
|
PFA RB RPMI SDS-PAGE TMB VHH |
Paraformaldehyde RNase B Roswell park memorial institute Sodium dodecyl sulfate polyacrylamide gel electrophoresis Tertramethylbenzidine Single variable domain on a heavy chain |
Reviewer 2 Report
Authors have written the manuscript very well. I just have few comments about the manuscript titled as "Expression and in vitro function of anti-breast cancer camelid single domain antibody VHH expressed in plants.
Please include error bars in Figures.
Author Response
Reviewer : 2
Authors have written the manuscript very well. I just have few comments about the manuscript titled as "Expression and in vitro function of anti-breast cancer camelid single domain antibody VHH expressed in plants.
Please include error bars in Figures..
→ The error bars are included with statistical analysis (Figs. 3 and 4).
→ The statistical analysis is described in M&M (Page 13 line 349~354)
Again, we are really appreciate to have an opportunity to revise our manuscript for publication in IJMS according to all reviewers comments and suggestions.
Round 2
Reviewer 1 Report
The authors adequately responded to all comments and performed all required modifications.